# Mechanical Properties, Wear Resistance, and Reliability of Two CAD-CAM Resin Matrix Ceramics

**DOI:** 10.3390/medicina59010128

**Published:** 2023-01-09

**Authors:** Ebele Adaobi Silva, Anselmo Agostinho Simionato, Adriana Cláudia Lapria Faria, Estevam Augusto Bonfante, Renata Cristina Silveira Rodrigues, Ricardo Faria Ribeiro

**Affiliations:** 1Department of Dental Materials and Prosthodontics, Ribeirao Preto School of Dentistry, University of Sao Paulo—FORP-USP, Sao Paulo 14040-904, Brazil; 2Department of Prosthodontics and Periodontology, Bauru School of Dentistry, University of Sao Paulo—FOB-USP, Sao Paulo 17012-901, Brazil

**Keywords:** CAD-CAM, dental materials, composite resin, ceramic, dental restoration

## Abstract

*Background and Objectives:* There are limited data regarding the behavior of resin matrix ceramics for current CAD-CAM materials. Further studies may be beneficial and can help clinicians planning to use these materials during prosthodontic rehabilitation. The aim of this study was to evaluate and compare the flexural strength and strain distributions, filler content, wear, and reliability of two resin matrix ceramic CAD-CAM materials. *Materials and Methods:* Two resin matrix ceramics, Ambarino High-Class (AH) and Vita Enamic (VE), were tested for flexural strength (n = 24), wear (n = 10), and reliability (n = 18). Thermogravimetric analysis was used to determine the percentage of filler by weight, and digital image correlation (DIC) was used for strain analysis in flexural strength test. Reliability of each resin matrix ceramic was compared after accelerated lifetime testing of crowns using a two-parameter Weibull distribution. Data of flexural strength, wear, and thermogravimetry were analyzed by independent t-tests with significance level at 5%. *Results:* The results of DIC analysis were analyzed by a qualitative comparison between the images obtained. The materials tested showed different flexural strength (*p* < 0.05) and strain distributions. The filler content was the same as informed by manufacturers. No difference was observed in the wear or reliability analysis (*p* > 0.05). The flexural strength of material AH was superior to VE, and the strain distribution was compatible with this finding. *Conclusions:* The two resin matrix ceramics tested showed similar behavior in wear and reliability analysis. Both can provide safe use for dental crowns.

## 1. Introduction

New technologies related to the manufacture of dental restorations are increasingly applied, particularly with greater accessibility to manufacturing methods, such as computer-aided design and manufacturing—CAD-CAM [1,2]. Modern materials and processes provide improvements over traditional methods, especially in terms of time consumption and predictability of results [3] s well as technical, biological, and aesthetic requirements [1]. Among the materials used for CAD-CAM, ceramics and composites are acknowledged as the most used. Ceramics have the advantages of esthetic appearance, biocompatibility, durability, and staining resistance, whereas composites have low abrasiveness to enamel antagonists, in addition to being easy to polish and repair [4]. Recently, new materials called resin matrix ceramics were developed to combine the characteristics of ceramics and composites.

Resin matrix ceramics, also called PICN (polymer-infiltrated ceramic network), are obtained from a porous pre-sintered ceramic network conditioned by a coupling agent and infiltrated with a polymer by capillary action [5,6]. Replacing the glass matrix of conventional ceramics with a polymer network improves the properties of flexural strength and strain to failure and may present mechanical behavior like natural dental tissue [1,5,6,7] and better machinability compared with ceramic materials [6]. However, different methodologies were used and high variability in the results are reported in the literature [1,5,6,8,9]. In addition, there are many combinations of polymers and fillers used by different manufacturers, which calls for caution in interpreting the results [5,8,10]. 

Failures can be introduced into restorations by chewing force, parafunctional habits, and adjustments made during laboratory or clinical procedures. Resin matrix ceramics are more tolerant to cracks than conventional ceramics due to the polymer matrix that reduces crack propagation [6]. Resin matrix ceramic materials, however, are affected by artificial aging, impairing their mechanical properties, such as Vickers hardness and flexural strength [7,11], in addition to presenting microcracks after aging, which is not observed with conventional ceramic materials [11]. Submitting resin matrix ceramic materials to conditions where artificial aging is present along with stress conditions, such as wear simulation, may help in the understanding of the behavior of the material under dynamic conditions. At the same time, reliability analysis allows for safety in the application of these materials.

There is limited data regarding the behavior of resin matrix ceramics for current CAD-CAM materials. Further studies may be beneficial and can help clinicians planning to use these materials during prosthodontic rehabilitation. The purpose of this study was to evaluate and compare the flexural strength and stress/strain distributions, filler content, wear, and reliability of two dental resin matrix ceramic CAD-CAM materials. The tested null hypothesis is that materials show similar flexural strength values and have no difference regarding the strain distribution, wear resistance, or reliability. The filler content of both resin matrix ceramics was investigated by thermogravimetry.

## 2. Materials and Methods

Two resin matrix ceramics (Table 1) for CAD-CAM were evaluated following the workflow shown in Figure 1.

### 2.1. Specimen Preparation

In this study, three-point bending test, wear test, and reliability analysis (compression strength and fracture resistance) were performed and required the preparation of specific specimens for each test. Bar-shaped, hemispherical, and canine-shaped crowns were obtained for the tests. The thermogravimetric analysis required amounts (12 mg) of fragmented material.

Bar-shaped specimens were obtained for the three-point bending test. The resin matrix ceramics were cut under irrigation with a diamond-cutting disc (Allied High Tech Products Inc, Racgo Dominguez, Compton, CA, USA) in a precision saw (Isomet 1000 Precision Saw, Buehler, IL, USA). Twenty-four specimens were obtained for each resin matrix ceramics with dimensions of 14.0 mm × 4.0 mm × 1.2 mm (n = 24) [7]. The specimens were polished with 320-, 400-, 600-, and 1200-grit sandpaper under water-cooling for 5 min each. Then, they were immersed in a distilled water ultrasonic bath (Plus 3LD, Ecel, Ribeirao Preto, Brazil) for 10 min and dried with paper towels. Dimensions of all specimens were checked using a digital caliper (Mitutoyo Sul Americana Ltd., Suzano, Brazil).

For two-body wear test, two types of patterns were prepared. Hemispherical specimens (n = 10) were obtained by the CAD-CAM method (Ceramill Motion 2, Amann Girrbach AG, Pforzheim, Germany) for each resin matrix ceramic (d = 5.0 mm), and twenty dental enamel antagonists from ten extracted third molars (approval by the Research Ethics Committee -CAAE: 21710619.9.0000.5419) that had their roots discarded and were sectioned in half in the mesiodistal direction (Isomet 1000 Precision Saw, Buehler, IL, USA) with a diamond-cutting disc (Allied High Tech Products Inc., Racgo Dominguez, Canada) were also obtained. The vestibular and lingual surfaces were flattened with 320-, 400-, 600-, and 1200-grit sandpaper under water-cooling. Then, the obtained samples were embedded in PVC rings (20 mm × 16 mm) using self-curing resin (VIPI Flash, VIPI, Pirassununga, Brazil). Hemispherical specimens and enamel samples were stored in distilled water at 37 °C for 7 days before testing [7].

For compression strength or fracture resistance tests, twenty-one maxillary canine-shaped crowns were obtained for each material using the CAD-CAM method (Ceramill Motion 2, Amann Girrbach AG, Pforzheim, Germany). They were cleaned with isopropyl alcohol and cemented with dual-cured resin self-adhesive cement (Megalink Auto TR, Odontomega, Ribeirao Preto, Brazil) in universal abutments with 3.3 mm diameter and 4.5 mm height (Singular Implants, Parnamirim, Brazil). The universal abutments were torqued (32 N.cm) to implants (Cone Morse Go Direct—3.5 mm × 11.5 mm, Singular Implants, Parnamirim, Brazil) embedded in polyurethane (F160, Axson, Cergy, France). After cementation, the sets were stored at 37 °C and 100% humidity for 24 h before testing.

### 2.2. Flexural Strength Test

Flexural properties were determined using a three-point bending test according to ISO 6872: 2008 [7,11], with a support span of 12.0 mm and a crosshead speed of 0.5 mm/min at room temperature (25 °C) using a universal testing machine (Biopdi, Sao Carlos, Brazil). The load was applied at the center of specimen until permanent deformation or rupture. The universal testing machine software calculated the flexural strength, following the formula:$$\sigma =\frac{3\mathrm{FL}}{2{\mathrm{bd}}^{2}}$$
where F is the load at the fracture point, L is the length of the support span, b is width, and d is thickness.

### 2.3. Digital Image Correlation Analysis

The specimens used for the Digital Image Correlation (DIC) analysis were the same that were used in three-point bending test. Samples were painted with a white background and sprayed with a fine layer of black paint to produce a surface that allows tracking and deformation analysis [12]. The DIC system (StrainMaster; LaVision Inc., Göttingen, Germany) includes two digital charged-coupled device cameras (Intense Imager, LaVision Inc.) to capture the images of deformation and a software (DaVis 10.1; LaVision Inc.) for image analysis to calculate the strains generated on specimen surface. Images of the painted surface were acquired at a frequency of 1 Hz during the three-point bending test. The images acquired were compared with the first image to calculate the displacement on the surface of the model [13]. Horizontal strains (Ԑ_xx_) were calculated on the basis of point displacements by DaVis 10.1 software.

### 2.4. Wear Test

Antagonistic dental enamel surfaces and hemispherical specimens were subjected to two-body wear testing apparatus developed in the Department of Dental Materials and Prosthodontics of Dental School of Ribeirao Preto, University of São Paulo [14,15]. Hemispherical specimens were fixed in a recipient that moved in a 5 mm linear course, with linear speed of 10 mm/s. The antagonistic dental enamel was fixed in adjustable vertical loading poles under a 20 N load. Each simulated chewing cycle included a downward vertical movement (occlusion), a 5 mm lateral movement (eccentric loading), and an upward vertical movement (disocclusion). Five dental enamel/hemispherical specimen assemblies were tested simultaneously. Then, 300,000 cycles at a frequency of 1 Hz were performed, simulating one year of masticatory function at the average human masticatory frequency [15,16].

At the beginning and at the end of the wear tests, hemispherical specimens’ profiles were traced using a profile projector (Nikon Profile Projector, 6C, Nikon, Tokyo, Japan) at 10× magnification on a transparent sheet. A device was used to standardize the samples position before and after the test. The height loss was measured using 0.01 mm digital caliper (Mitutoyo Sul Americana Ltd., Suzano, Brazil) and represents the hemispherical specimen height loss.

For the evaluation of mass loss at hemispherical specimens, the mass was determined at the beginning and at the end of the tests using an analytical balance (Bel Engineering, Monza, Italy) with a resolution of 0.1 mg. Mass loss was determined by subtracting the final values of the initial values of the mass. 

### 2.5. Filler Content

Thermogravimetric analysis (TG) was used to determine the percentage of fillers by weight using a Thermogravimetric Analyzer (TGA/DSC SDTQ600, TA Instruments—New Castle, DE, USA). Three fragmented and grinded specimens of each material (12 ± 1 mg) were heated at a rate of 10 °C min^−1^ up to 700 °C under nitrogen atmosphere. During the process, the polymer was eliminated, and the residue represented the amount of fillers of each material. The Thermogravimetric Analyzer software was used to calculate the percentage weight loss corresponding the organic part [17].

### 2.6. Mechanical Testing and Reliability Analysis

Three maxillary canine-shaped crowns (n = 3) of each group underwent single load-to-failure (SLF) testing at 30° inclination using 1000 KgF load cell and crosshead speed of 1 mm/min in a universal testing machine (Biopdi, Sao Carlos, Brazil) [18,19]. On the basis of mean SLF values, crowns were assigned to three step-stress profiles (Figure 2) used in step-stress accelerated life testing (SSALT) following 3:2:1 ratio and designated as mild (n = 9), moderate (n = 6), and aggressive (n = 3). This methodology was used to evaluate and compare the reliability of ceramic materials under accelerated lifetime fatigue test [20]. The maximum load used in the aggressive step corresponded to 60% of the average values obtained in the SLF. 

Eighteen crowns from each group were cycled on a fatigue testing equipment (Biocycle, Biopdi, Sao Carlos, Brazil). Simultaneously with mechanical cycling, the samples were subjected to thermal cycling, with a temperature varying between 5 °C and 55 °C, with an immersion time of 40 s at each temperature. The first level of each profile was 50,000 cycles, increasing by 10,750 cycles at the end and beginning of the next level. The equipment was programmed with progressive loads at each level. The mild and moderate profiles started with a load of 80 N, and the aggressive profile started with a load of 100 N. The increased load at each level varied according to each profile and finished in 280 N for all. On the basis of step-stress distribution of failures, probability Weibull curves were calculated (Synthesis 9, Alta Pro, Reliasoft, Tucson, AZ, USA) using a power law relationship for damage accumulation [20]. Reliability for a mission of 50,000 cycles at 100 N and 150 N (90% two-sided confidence interval) was calculated for comparison between the groups. Weibull analysis provides the beta value (β), which describes the behavior of failure rate over time. Beta values less than 1 (β < 1) indicate failure rate decreases over time and is associated with early failures. Beta values of approximately 1 (β~1) determine constant failures rates over time and is associated with random failures. Beta values greater than 1 (β > 1) indicate that the failure rate increases over time and is linked to fatigue damage. According to β < 1 for the tested resin matrix ceramics, the contour plot of Weibull probability was calculated using final stress for group failure or survival (90% confidence intervals).

### 2.7. Statistical Analysis

Data of flexural strength, wear, and thermogravimetry were analyzed by independent t-test with significance level at 5% using the software SPSS 20.0 (IBM SPSS Software, IBM Corporation). The results of DIC analysis were analyzed by a qualitative comparison between the images obtained. Frequency of failures at each resin matrix ceramic was compared by Weibull curves.

## 3. Results

The mean values of flexural strength and wear (height loss and mass loss) are presented in Table 2. According to the results, material AH obtained significantly higher values of flexural strength (*p* < 0.05) than material VE. The height loss (*p* = 0.671) and mass loss (*p* = 0.241) of both materials was not significantly different for either of the two resin matrix ceramics.

The results of DIC represent qualitatively the distribution of horizontal strain on the specimen during the three-point bending test (Figure 3 and Figure 4). Cold colors (blue to green) represent compressive strains, and warm (red to green) colors represent tensile strains. The horizontal stress values indicated by the color scale agree with the results obtained during the three-point bending strength test; the image relative to material AH shows higher microstrain (µS) values than material VE, which indicates the need for a higher load applied until the failure.

The thermogravimetric analysis showed that the VE material (86%) had a significantly (*p* < 0.05) greater filler content than material AH (70%), and material AH had a greater amount (*p* < 0.05) of polymeric material.

The graph and summary statistics for level probability derived from the Weibull step-stress with 150 N usage stress are presented in the Figure 5 and Table 3. The mean values of β (confidence interval limits) derived from the Weibull calculations of the probability of use (probability of failure versus number of cycles) were 0.5 for AH and 0.38 for VE. The resulting β values indicated that, regardless of the stage stress level at which the samples were fatigued, failures were associated with load (level of stress) rather than accumulation of fatigue damage. Thus, the Weibull distribution probability was determined using failure load during fatigue.

The Weibull modulus (m) was 6.41 (upper limit: 8.66; lower limit: 4.74) for AH and 9.43 (upper limit: 12.73; lower limit: 6.99) for VE. The characteristic resistance values (η) were 211.25 N (upper limit: 225.04 N; lower limit: 198.31 N) for group AH and 222.64 N (upper limit: 232.83 N; lower limit: 212.89 N) for the VE group. The contour plot (characteristic strength vs. Weibull modulus) exhibits this information graphically and detects that these datasets are from similar populations (*p* > 0.05) on the basis of the overlapping confidence limits (Figure 5). The calculated reliability was similar for the two materials at any given mission (*p* > 0.05). The behavior of the groups is closer in the 100,000 cycles mission under a load of 150 N. With the load in 200 N, for a mission of 100,000 cycles, there was a decrease of 62.16% for AH and 48.89% for VE when compared with the load of 150 N. With the increase in the number of cycles to 150,000 and loading retained at 200 N, there was a decrease of 25% for group AH and 13.04% for group VE, when compared with the mission of 100,000 cycles, with the VE material for the longest mission in 200 N being the least affected.

## 4. Discussion

The null hypothesis was partially rejected because the two resin matrix ceramics tested showed a significant difference in flexural strength, which was higher for material AH, and showed different strain distributions in the DIC analysis. The thermogravimetry results showed differences in the content of fillers and polymer between AH and VE, which explain different behaviors in some of the tests performed.

Fracture toughness is reported as one of the main properties associated with clinical performance of dental materials, indicating their ability to resist crack propagation and catastrophic failure, especially in brittle materials [5,15]. The propensity to fracture in restorations when using ceramic materials and to fracture in the dental substrate when using resin-based materials is observed, which leads to concerns about fracture resistance. Furthermore, ceramic materials are susceptible to damage from machining, especially in small thicknesses, such as on restoration margins [21]. The DIC analysis highlights the strain behavior during the three-point bending test [1]. By qualitatively analyzing the strain distribution in the ceramic bars during the bending test through digital image correlation, the behavior of the materials was compatible with the results discussed above. Material AH showed higher flexural strength values and compatible results for DIC, with lower distribution of compressive and tensile strain on the bar surface and higher microstrain (µS) values obtained until bar rupture. One of the limitations of this method is the capture of images until the bar failure, which happens suddenly and does not allow full interpretation of crack propagation.

Resin matrix ceramics are composed of a polymeric matrix and inorganic filler reinforcement particles [5]. Microstructurally, there is a similarity with composites and ceramics [5]. If the reinforcement particles are not well dispersed or bonded to the matrix material, they can serve as limiting factors of resistance [3] and can contribute facilitating crack development and failures [3,4]. In the present study, the analysis of the structures of the two materials was not performed, but this may be one of the explanations for the significantly lower flexural strength of VE in comparison with AH. Hampe et al., 2019 [3] performed a fractographic analysis that showed different topographic fracture patterns and differences in fracture toughness between VE and AH, where AH showed higher values in fracture toughness, even after thermocycling [3], which supports the results of this study. Another factor that can explain the differences between the two resin matrix ceramics is the polymer matrix used, which, according to the information provided by the manufacturers, is different. Bis-GMA, TEGDMA, and UDMA have different structure and properties; Bis-GMA enhances the flexural modulus and decreases the flexural strength of the polymer, and UDMA influences the elasticity by decreasing flexion modulus [22]. Different combinations of these monomers provide different properties to the polymer matrix by influencing the amount of double bonds and degree of conversion [23,24].

The three-point bending test is commonly used to evaluate the flexural strength of restorative or prosthetic materials using bar-shaped specimens [4,8,24]. International standards for flexural strength testing of resin-based (ISO 4049) and ceramic-based (ISO 6872) materials require that specimens have a length and depth that allow for a ≥10 ratio. However, CAD-CAM blocks have reduced dimensions, especially if they are intended for single restorations. Choi et al., 2019 [22] stated that there is discrepancy in the flexural strengths found in studies [4,7,8,25] that used bars with a dimension 14.0 mm × 4.0 mm × 1.2 mm with a support span of 10 mm to 12 mm, leading to incorrect interpretations of the mechanical properties. Several factors originating from the geometry of the bars can affect the results [22]. This may explain the difference in values obtained for flexural strength from the values provided by the manufacturers. The CAD-CAM blocks used did not allow for the milling of specimens with larger dimensions, suggesting that a new test protocol could be developed for accurate measurement of flexural properties of these materials. The ISO 6872:2008 standard recommends an average flexural strength above 300 MPa for using a material as a three-unit anterior fixed dental prosthesis [2]. The results obtained for the resin matrix ceramics used in this study lead to recommendation of these materials for anterior or posterior single crowns.

Methods, such as thermogravimetry, can be used to verify the polymer content and filler content of a material [17]. The thermogravimetric analysis was applied on this study to analyze the behavior of the tested materials to measure filler content. Polymer-based materials show lower hardness than ceramic materials, facilitating their milling with less edge chipping and tool wear; however, the large concentration of filler content makes them behave as brittle materials [3]. Filler content, morphology, and distribution of filler particles may have an impact on mechanical properties [17]. The results of this study showed that the higher filler content of the VE material caused strain distribution and behavior of this material during DIC analysis similar to a brittle material, and the lower concentration of filler content in AH caused this material to show different behavior of the VE in this analysis. The images obtained show that the behavior presented by the materials is compatible with the flexural strength results found in this study.

Wear can happen through mechanisms such as tooth-brushing, food wear, load on the occlusal area, and erosion, which leads to exposure of the polymer matrix and filler particles [26]. Stable occlusal contacts are important for the longevity of a restoration. With continuous use, the material used in a restoration must have wear resistance to maintain the cusp height and vertical dimension of occlusion [27,28]. Although in vitro studies cannot fully reproduce the oral environment, it is possible to simulate a similar environment to assess the clinical performance of the material [29]. In this study, one year of masticatory function [17] was simulated, with the specimens immersed in distilled water at 37 °C, performing occlusion, sliding, and lift-off cycling, having tooth enamel as antagonist, with no difference between the resin matrix ceramics tested. It was observed that, numerically, material AH had greater height loss, and material VE had greater mass loss, without difference. This can be explained by the density of the materials, which leads to greater mass loss for VE (2.1 g/cm^3^), even with less height loss than AH (1.89 g/cm^3^). The results of the thermogravimetric analysis show that material AH has a polymer content greater than VE, which may also have contributed to the wear results obtained. The wear of a material is a slow, continuous, and multifactorial process [27,30], which suggests that longer periods of use could promote cumulative significant wear among the resin matrix ceramics tested. One of the limitations of the wear analysis in this study is the absence of roughness evaluations on the surfaces of the hemispherical specimens and on the enamel antagonists, which could contribute to the interpretation of the results found.

Restoration failures can be better understood by dynamic fatigue studies. Cyclic loadings simulating the function under alternating thermal and mechanical conditions allows some insight regarding potential clinical performance [9,19]. Analyzing failure probability with the Weibull distribution along with failure modes provide a reasonable indication of the clinical performance of the material [31]. The Weibull modulus (m) is a unitless parameter that describes the variation of the results as a function of flaw distribution within a material [20]. Although the higher the module, the smaller the variation and the greater the reliability [31], values were not different between groups. The characteristic strength value (η) denotes 63.2% of specimens with an expectation of failure [32], and this was not significantly different between materials. There is still no standardization regarding the classification of these new materials, as both are said to be hybrids; however, their compositions present differences (% ceramic and polymer) which, as observed, led to different behaviors against certain stimuli [33,34,35]. An in vitro test cannot replicate all biological factors in the oral environment, and clinical conditions may lead to different results due to the added effect of stress and the complex environment.

The present study has limitations, such as the failure to carry out a microstructural analysis that could help clarify the distribution of the different phases of the material after producing the restorative piece. Another aspect is the issue of the dimensions of the bars used for the analysis of flexural strength and the correlation of digital images. The available blocks did not allow the dimensions indicated by the ISO 6872:2018 standard and, therefore, care must be taken when comparing these results with those in the literature. Thus, new laboratory and, especially, clinical studies should be carried out to obtain a more complete evaluation of resin matrix ceramics. Even so, resin matrix ceramics are materials with very good potential for use in the production of indirect restorations.

## 5. Conclusions

Within the limitations of this study, it was concluded that:flexural strength of material AH was higher than that of VE, and the strain distribution observed by DIC was compatible with that;the two resin matrix ceramics showed similar behavior in the wear resistance test;the two resin matrix ceramics had similar results for the reliability analysis;the results of this study lead to the recommendation of both materials for anterior and posterior single crowns.

## Figures and Tables

**Figure 1 medicina-59-00128-f001:**
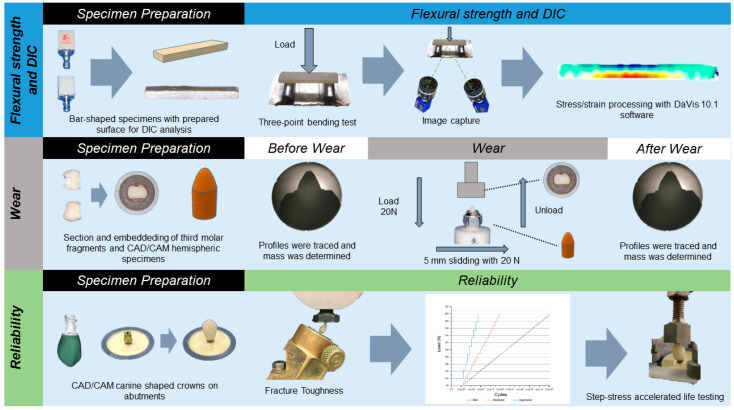
Experimental design representing the workflow adopted for the performed analyses.

**Figure 2 medicina-59-00128-f002:**
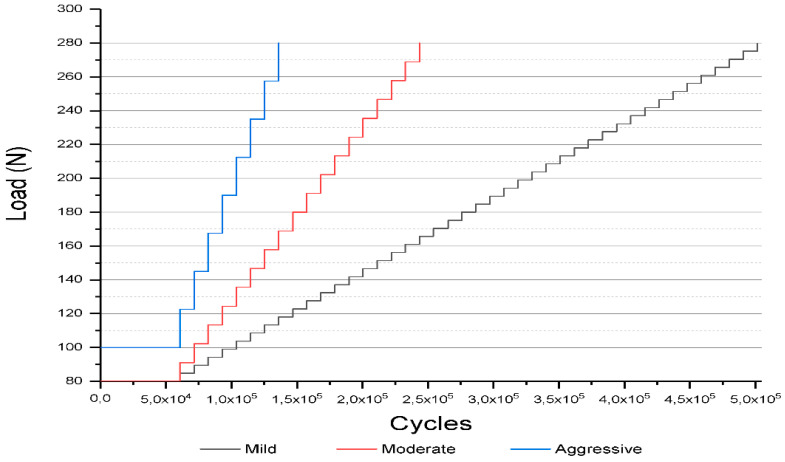
The step-stress profiles utilized for accelerated life testing (SSALT) on the basis of the mean value of single load to failure (SLF).

**Figure 3 medicina-59-00128-f003:**
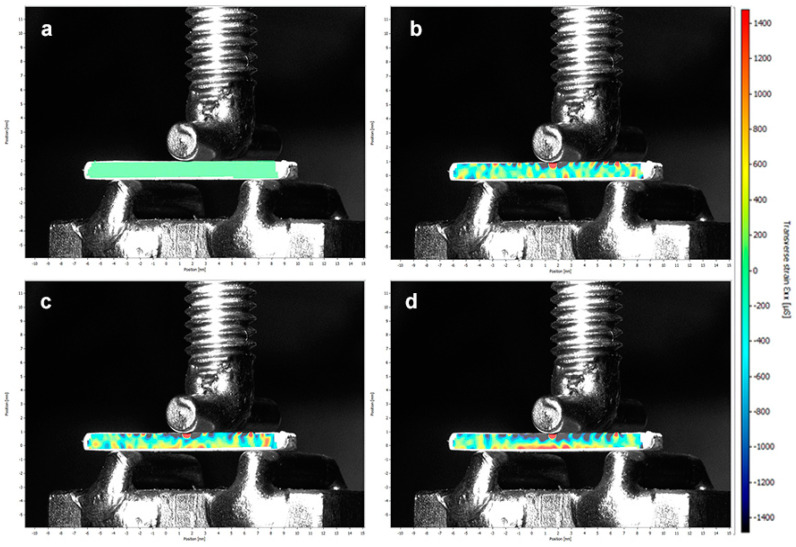
The sequence of photographs obtained in the DIC analysis of the material VE: (**a**) without strain concentration; (**b**) when the test starts; (**c**) with more load on specimen the strain concentration increases; and (**d**) the last photograph before failure.

**Figure 4 medicina-59-00128-f004:**
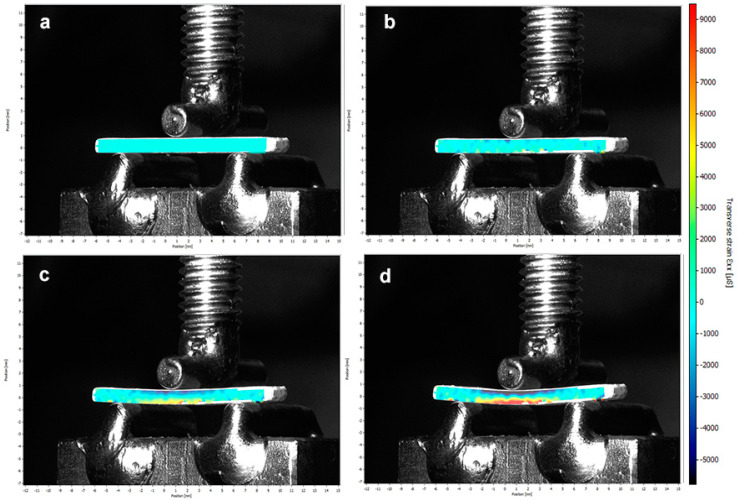
The sequence of photographs obtained in the DIC analysis of the material AH: (**a**) without strain concentration; (**b**) when the test starts; (**c**) with more load on specimen the strain concentration increases; and (**d**) the last photograph before failure.

**Figure 5 medicina-59-00128-f005:**
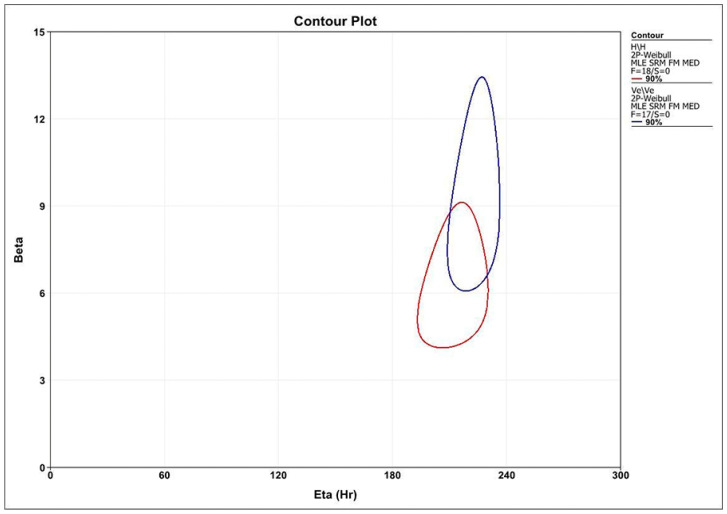
Contour plot: Weibull modulus vs. characteristic resistance (the red contour shows the results for the AH material and the blue contour for the VE material).

**Table 1 medicina-59-00128-t001:** Composition and properties of CAD/CAM blocks examined as published by manufacturers.

Type	Brand	Code	Composition	Filler Mass %	Flexural Strength (MPa)
Resin matrix ceramic	Vita Enamic^®^	VE	UDMA, TEGDMA (Monomer); Feldspar ceramic enriched with aluminum oxide (Filler)	86	150–160
Resin matrix ceramic	Ambarino^®^ High-Class	AH	Nanocharges, BDDMA, Bis-GMA, UDMA (Monomer); Strontium boroaluminosilicate glass (Filler)	70	191

Abbr.: UDMA: urethane dimethacrylate; TEGDMA: triethylene glycol dimethacrylate; Bis-GMA: bisphenol A diglycidylether methacrylate; BDDMA: 1,4 Butanediol dimethacrylate.

**Table 2 medicina-59-00128-t002:** Mean values and standard deviation of flexural strength and wear.

	VE	AH	*p*-Value
Flexural Strength (MPa)	104.54 (14.07) a	123.24 (18.35) b	*p* < 0.05
Height Loss (µm)	560.125 (300.63)	630.75 (348.77)	0.671
Mass Loss (mg)	2.2625 (2.1573)	1.2625 (0.8193)	0.241

Different lowercase letters indicate statistical difference between columns.

**Table 3 medicina-59-00128-t003:** Reliability for missions of 50,000 cycles according to load; upper and lower limits (90% confidence interval).

Material	AH			VE		
	Lower Limit	Reliability	Upper Limit	Lower Limit	Reliability	Upper Limit
100k/150 N	0.56	0.74 a	0.86	0.75	0.90 a	0.96
100k/200 N	0.14	0.28 b	0.43	0.26	0.46 b	0.63
150k/200 N	0.08	0.21 b	0.38	0.20	0.40 b	0.60
Beta (β)	0.5			0.38		
Characteristic Strength (MPa)	448.19 (13.10)			462.79 (52.57)		
Weibull modulus (*m*)	4.74	6.41	8.66	6.99	9.43	12.73

Different lowercase letters indicate significant difference (*p* < 0.05).

## Data Availability

Data supporting the results of this study are available in the article and can be requested from the corresponding author.

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
