# Peer review of "Mechanical Properties, Wear Resistance, and Reliability of Two CAD-CAM Resin Matrix Ceramics"

_medicina, 2023, doi:10.3390/medicina59010128_

Round 1

Reviewer 1 Report

I thank the authors for this detailed and painstaking work. I am sure it will contribute to the literature. However, I believe making the following minor corrections will improve the quality of the manuscript.

Abstract: Well written.

Introduction: Well written.

Materials and methods: Why were the samples mentioned in lines 102 and 103 soaked in water for 7 days? Is there any literature support on this? If there is, please add it to the references.

Please explain the phrase kept at 100% humidity on lines 111 and 112. Give more details.

Results: Well written.

Discussion and conclusion: Well written.

References: References 16, 18, 23, 31, 35, and 37 are outdated. Please replace with new ones.

Author Response

Response to Comments and Suggestions for Authors

I thank the authors for this detailed and painstaking work. I am sure it will contribute to the literature. However, I believe making the following minor corrections will improve the quality of the manuscript.

Response: We appreciate the comments and care in evaluating the manuscript.

Abstract: Well written.

Introduction: Well written.

Materials and methods: Why were the samples mentioned in lines 102 and 103 soaked in water for 7 days? Is there any literature support on this? If there is, please add it to the references.

Please explain the phrase kept at 100% humidity on lines 111 and 112. Give more details.

Response: Added the requested reference that discusses the importance of wet storage for testing.

Results: Well written.

Discussion and conclusion: Well written.

References: References 16, 18, 23, 31, 35, and 37 are outdated. Please replace with new ones.

Response: The references were deleted and the final list was reorganized accordingly.

Reviewer 2 Report

The study is genuine and very interesting. The manuscript is well-written, however the authors are required to address the following points to improve the overall quality of the manuscript:

- The authors should add a short background statement in the abstract and mention the current research gap briefly.

- Figure 1 is excellent however the size should be increased to improve the visibility of its contents.

- How did the authors determine the sample size? Please clarify this matter in the text.

- Please revise the Glossary of Prosthodontic Terms for reporting scientific terms. For example, the word "superior" should be replaced by "maxillary".

- The discussion part should include the study limitations and directions for future research.

- The conclusion section should be expanded to show the significant outcomes of the study and summarized in bullet points.

Author Response

Response to Comments and Suggestions for Authors

The study is genuine and very interesting. The manuscript is well-written, however the authors are required to address the following points to improve the overall quality of the manuscript:

Response: We appreciate the comments and care in evaluating the manuscript.

- The authors should add a short background statement in the abstract and mention the current research gap briefly.

Response: Fixed as per suggestion.

- Figure 1 is excellent however the size should be increased to improve the visibility of its contents.

Response: We increased the size of the most important Figures in the study, as suggested.

- How did the authors determine the sample size? Please clarify this matter in the text.

Response: We had some limitations due to the availability of material that was given to us for the study.

For the flexural strength test, as far as possible, we used 24 samples, therefore within the range suggested by the ISO 6872 standard, which establishes a minimum of 10 and an ideal of 30 samples.

In these samples, concomitantly, the images of the Digital Image Correlation (DIC) were acquired.

For the wear test we used 10 samples, limited both by the material and by the availability of human teeth in the Faculty's Teeth Bank.

For the reliability test, we followed the norm established in the SSALT methodology (Step-stresses accelerated lifetime testing), which, based on the compressive strength test of 3 samples, establishes the range of forces to be applied. Afterwards, 18 samples are divided into three groups: light loading with 9 samples; moderate loading with 6 samples; and severe loading with 3 samples.

For thermogravimetry, the request is for the amount by weight of the material (12 mg), which was followed.

- Please revise the Glossary of Prosthodontic Terms for reporting scientific terms. For example, the word "superior" should be replaced by "maxillary".

Response: Corrections made accordingly the Glossary of Prosthodontic Terms as requested.

- The discussion part should include the study limitations and directions for future research.

Response: A new paragraph was added as requested.

- The conclusion section should be expanded to show the significant outcomes of the study and summarized in bullet points.

Response: Conclusions rewritten as suggested.